# Subspecies Niche Specialization in the Oral Microbiome Is Associated with Nasopharyngeal Carcinoma Risk

Justine W. Debelius,[a*] Tingting Huang,[a,b] Yonglin Cai,[c,d] Alexander Ploner,[a] Donal Barrett,[a] Xiaoying Zhou,[e,f] Xue Xiao,[g] Yancheng Li,[c,d] Jian Liao,[h] Yuming Zheng,[c,d] Guangwu Huang,[g] Hans-Olov Adami,[a,i] Yi Zeng,[j] Zhe Zhang,[g] Weimin Ye[a]

aDepartment of Medical Epidemiology and Biostatistics, Karolinska Institutet, Stockholm, Sweden
bDepartment of Radiation Oncology, The First Affiliated Hospital of Guangxi Medical University, Nanning, People's Republic of China
cDepartment of Cancer Prevention Center, Wuzhou Red Cross Hospital, Wuzhou, People's Republic of China
dWuzhou Health System Key Laboratory for Nasopharyngeal Carcinoma Etiology and Molecular Mechanism, Wuzhou, People's Republic of China
eLife Science Institute, Guangxi Medical University, Nanning, People's Republic of China
fKey Laboratory of High-Incidence-Tumor Prevention & Treatment (Guangxi Medical University), Ministry of Education, Nanning, People's Republic of China
gDepartment of Otolaryngology-Head & Neck Surgery, First Affiliated Hospital of Guangxi Medical University, Nanning, People's Republic of China
hCangwu Institute for Nasopharyngeal Carcinoma Control and Prevention, Wuzhou, People's Republic of China
iClinical Effectiveness Research Group, Institute of Health, University of Oslo, Oslo, Norway
jState Key Laboratory for Infectious Diseases Prevention and Control, Institute for Viral Disease Control and Prevention, Chinese Center for Disease Control and Prevention, Beijing, People's Republic of China

Justine W. Debelius, Tingting Huang, and Yonglin Cai contributed equally to this work. Authorship order was determined based on seniority.

Yi Zeng, Zhe Zhang, and Weimin Ye contributed equally to this work. Authorship order was established by consortium policy.

**ABSTRACT** Oral health and changes in the oral microbiome have been associated with both local and systemic cancer. Poor oral hygiene is a known risk factor for nasopharyngeal carcinoma (NPC), a virally associated head and neck cancer endemic to southern China. We explored the relationship between NPC and the oral microbiome using 16S rRNA sequencing in a study of 499 NPC patients and 495 population-based age and sex frequency-matched controls from an area of endemicity of Southern China. We found a significant reduction in community richness in cases compared to that in controls. Differences in the overall microbial community structure between cases and controls could not be explained by other potential confounders; disease status explained 5 times more variation in the unweighted UniFrac distance than the next most explanatory variable. In feature-based analyses, we identified a pair of coexcluding *Granulicatella adiacens* amplicon sequence variants (ASVs) which were strongly associated with NPC status and differed by a single nucleotide. The *G. adiacens* variant an individual carried was also associated with the overall microbial community based on beta diversity. Co-occurrence analysis suggested the two *G. adiacens* ASVs sit at the center of two coexcluding clusters of closely related organisms. Our results suggest there are differences in the oral microbiomes between NPC patients and healthy controls, and these may be associated with both a loss of microbial diversity and niche specialization among closely related commensals.

**IMPORTANCE** The relationship between oral health and the risk of nasopharyngeal carcinoma (NPC) was previously established. However, the role of oral microbiome has not been evaluated in the disease in a large epidemiological study. This paper clearly establishes a difference in the oral microbiomes between NPC patients and healthy controls which cannot be explained by other confounding factors. It furthermore identifies a pair of closely related coexcluding organisms associated with the disease, highlighting the importance of modern methods for single-nucleotide resolution in 16S rRNA sequence characterization. To the best of our knowledge, this is

Address correspondence to Weimin Ye, weimin.ye@ki.se.

* Present address: Justine W. Debelius, Center for Translational Microbiome Research, Department of Microbiology, Tumor and Cancer Biology, Karolinska Institutet, Stockholm, Sweden.

Endemic Nasopharyngeal Carcinoma is associated with the co-exclusion of two closely related Granulicatella adiacens variants and a loss of diversity in the oral microbiome in a large epidemiological study.

one of the first examples of cancer-associated niche specialization of the oral microbiome.

**KEYWORDS** 16S sequencing, *Granulicatella adiacens*, cancer, case-control study, microbiome, nasopharyngeal carcinoma, oral microbiome

The microbiome, including the oral microbiome, is emerging as an important factor in cancer and carcinogenic processes. *Helicobacter pylori* is perhaps the best-known example of an oncogenic bacteria; sequencing studies have implicated a new set of organisms, including *Fusobacterium nucleatum* in colorectal cancer, although cancer-associated differences in the microbiome have been seen across a wide variety of tumor types (1). Proposed mechanisms have included genotoxicity, immune regulation, and the production of oncogenic metabolites (2). Furthermore, differences in the local microbiome have been shown in cancer-associated viral infections, including chronic hepatitis infection and human papillomavirus (3–6). The microbiome is known to contribute to and modulate infection, persistence, reactivation, and transmission of oncogenic viruses, while viral infection may contribute to immune regulation of the microbiome (7).

Nasopharyngeal carcinoma (NPC) is a virally associated cancer endemic to areas such as southern China, Southeast Asia, the Arctic, and the Middle East/North Africa. In these regions, the incidence rate of NPC is more than 20 times higher than in the rest of the world (8). Infection with the Epstein-Barr virus (EBV) is the most widely accepted and extensively studied etiological factor, although its prevalence in the adult population worldwide approaches 90% (9, 10). The geographic and cultural differences associated with NPC incidence suggest that both genetic susceptibility and environmental/lifestyle factors, such as cigarette smoking and salted fish consumption, contribute to carcinogenesis (9, 11–13).

Several of these NPC risk factors may affect oral health and the oral microbiome (14–17). Poor oral hygiene is a risk factor for NPC, and a recent small-scale study suggested differences in the oral microbiomes between NPC patients and controls prior to radiation treatment (18, 19). However, it failed to fully address the way in which an NPC-related risk factor might confound this relationship, especially with regard to smoking. Therefore, we carried out a population-based case-control study in an area of endemicity of southern China (20). We analyzed microbial communities from 499 untreated incident NPC cases and 495 age and sex frequency-matched controls and addressed the relationship between NPC status and the oral microbiome adjusted for potential confounders.

## RESULTS AND DISCUSSION

Study participants were recruited from the Wuzhou region in Southern China between 2010 and 2014 as part of a large population-based case-control study (20). Saliva was collected during interviews. The oral microbiome was characterized using the V3-4 hypervariable region of the 16S rRNA gene; paired-end sequences were denoised to amplicon sequence variants (ASVs). ASVs provide a molecular barcode with single-nucleotide resolution to distinguish different organisms. After sequencing and denoising, there were 1,066 subject samples which had sufficiently high-quality sequences and metadata to be retained for analysis (see Fig. S1 in the supplemental material). Preliminary investigation suggested the microbiota of a small number of former smokers were highly heterogenous ($n = 72$; 33 cases, 39 controls) (see Fig. S2). We excluded former smokers from the final analysis, retaining 994 individuals (see Table S1; Fig. S1).

The ASV table from the 994 individuals included 5,824 ASVs from 20 classes, 66 families, and 125 genera. Superficially, the communities were similar to previously reported oral microbiota in Chinese adults (21, 22). The most abundant genera were *Streptococcus* (class *Bacilli*; mean relative abundance, 28.1%; 25% to 75% interquartile region [IQ], 17.4% to 36.1%), *Neisseria* (class *Betaproteobacteria*; mean, 19.3%; IQ, 9.7%

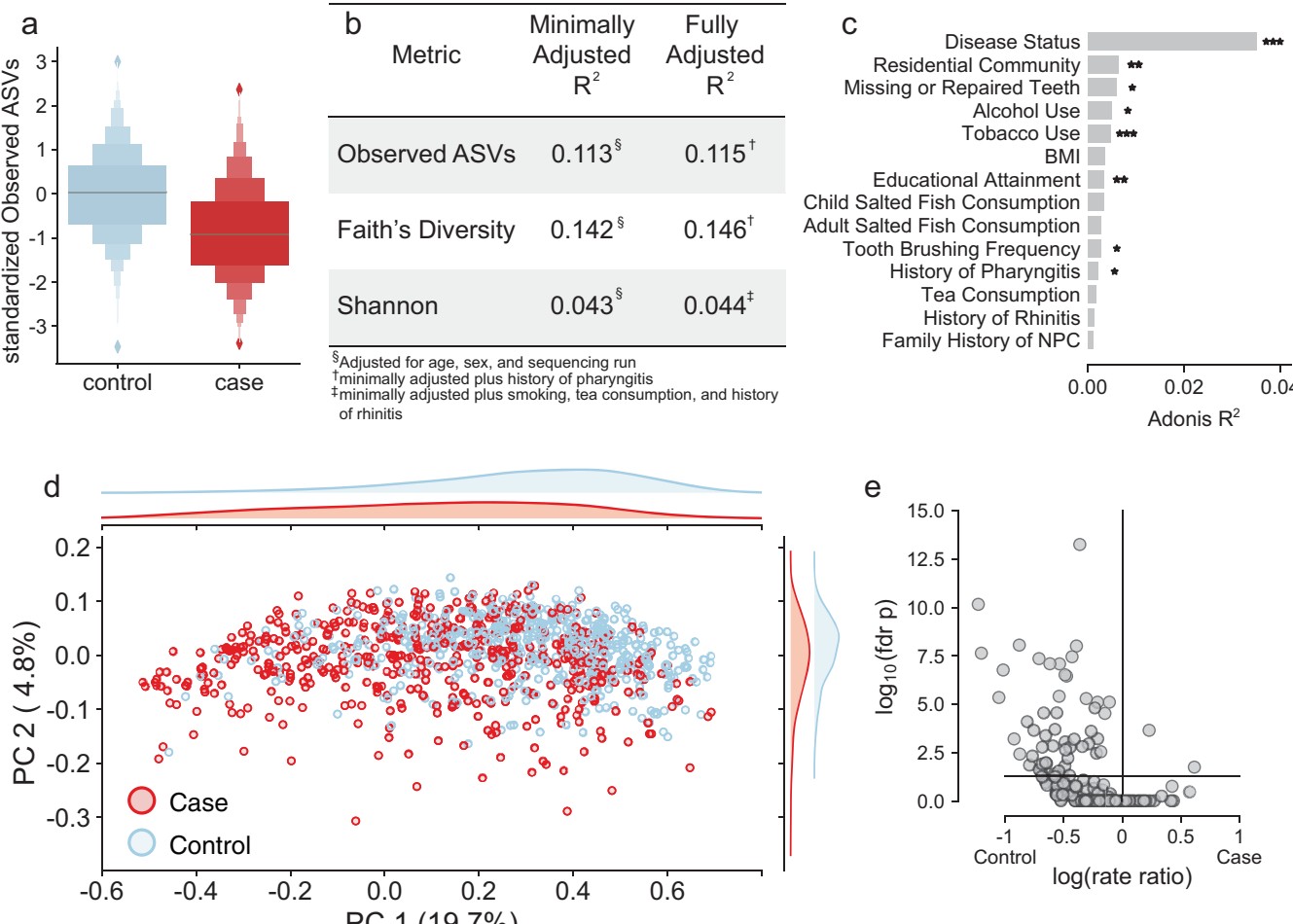

FIG 1 The oral microbiome differs between patients with nasopharyngeal carcinoma (NPC) and healthy controls. (a) NPC cases have significantly lower microbial richness than controls ($P < 1 \times 10^{-12}$) based on the z-normalized observed ASVs. The horizontal lines in the boxes represent the medians, the large boxes are the interquartile regions, and increasingly smaller boxes are the upper and lower eighths, sixteenths, etc., in the data, reflecting the distribution. (b) This difference is reflected in the proportion of variance explained by $R^2$ from a multivariate regression model. (c) Adonis testing with a model adjusted for age, sex, and sequencing run shows that for unweighted UniFrac distance, NPC diagnosis has more than five times the explanatory power of the next most important variable, residential community. For 9,999 permutations: ***, FDR-adjusted $P < 0.001$; **, $P < 0.01$; *, $P < 0.05$. (d) Principal-coordinates analysis (PCoA) of unweighted UniFrac distance shows separation between cases and controls along PC1 and PC2. At the top and right are the density distributions along each axis. The axes are labeled with the variation they explain. In unweighted UniFrac, PC1 explains 19.7% and PC2 explains 4.8% of the variation. (e) A volcano plot of the log prevalence ratio for disease status versus the log $P$ value reflects reduced diversity. The horizontal line indicates significance at a Benjamini-Hochberg corrected $P$ value of less than 0.05.

to 27.7%), *Prevotella* (class *Bacteroidia*; mean, 12.1%; IQ, 6.0% to 16.9%), *Veillonella* (class *Clostridia*; mean, 8.8%; IQ, 4.0% to 11.7%), and *Haemophilus* (class *Gammaproteobacteria*; mean, 5.5%; IQ, 2.8% to 7.4%).

We aimed to address the relationship between NPC and the oral microbiome, adjusted for potential confounders. As a result, we looked for factors which might affect the oral microbiome at a community level. Our primary confounders included oral hygiene and health, tobacco use, family history of NPC, and tea consumption (11, 12, 14–18, 23, 24). We also considered a history of oropharyngeal inflammation, the region where an individual lived, and alcohol use as covariates primarily expected to affect the microbiome, as well as salted fish consumption, which is primarily seen as a risk factor for NPC (13, 25, 26).

When comparing alpha diversity between cases and controls, we found that NPC cases showed significantly fewer overall ASVs, reduced phylogenetic diversity, and reduced Shannon diversity compared to those of controls (rank sum $P < 0.001$) (Fig. 1a; Table S2); these findings did not change after adjusting for covariates which were

significantly associated with alpha diversity (Fig. 1b). Hence, this suggests that patients newly diagnosed with NPC have lower overall microbial diversity than healthy controls. Our results agree with a small study of the oral microbiome in NPC patients ($n = 90$), which also found reduced alpha diversity (19). Unlike other body sites, there is no clear relationship between salivary microbiome richness and the health of the microbial community.

Similarly, when comparing global community patterns (beta diversity) via Adonis models minimally adjusted for sex, age, and sequencing run, we found significant differences between NPC cases and controls based on unweighted UniFrac, weighted UniFrac, and Bray-Curtis distances (false discovery rate [FDR] $P < 0.001$, 9,999 permutations) (Fig. 1c and d; see also Fig. S3a and b) (27–29). Compared to the potential confounders in the same setting, NPC status was the strongest explanatory factor for unweighted UniFrac distance, more than five times the effect size of the next strongest variable, as well as the second-strongest factor for weighted UniFrac and Bray-Curtis distances, just after tobacco use. There was no statistically significant difference in dispersion between cases and controls in any metric, supporting the idea that the differences are due to consistent differences between cases and controls ($P > 0.55$, 999 permutations) (Fig. 1d). Significance persisted in more fully adjusted Adonis models, including potential confounders with robust differences in community patterns and after stratification for tobacco use.

These findings establish that NPC status and smoking are strongly associated with differences in the oral microbiome in our population; the association with NPC is especially strong with regard to presence and absence of organisms (as emphasized by unweighted UniFrac distance) but second only to smoking with regard to abundances (as captured by weighted UniFrac and Bray-Curtis distances). We found no evidence that these associations are driven by community heterogeneity. Our results are, however, robust under adjustment for observed confounders, including smoking, residential community, and the number of missing or repaired teeth. In the case of the unweighted UniFrac distances, it is also unlikely to be the result of confounding by unobserved factors due to the crushing dominance of the signal for NPC status. Since we recruited incident patients prior to chemotherapy or radiotherapy treatment, it is also implausible that the observed differences in microbiome composition are treatment related (19, 20). We were unable to obtain information about previous antibiotic treatment, and it was not an exclusion criterion for this study, which is a major limitation in this analysis. Antibiotic treatment could be associated with decreased alpha diversity, although there is evidence to suggest that the oral microbiome is more robust to antibiotic use than that at other body sites and recovery from antibiotic use can be seen within 1 month of use (30). Even if we assume NPC patients were more likely to use antibiotics prior to enrollment, it seems unlikely that there was sufficiently widespread use to explain the large drop in richness we observed. Therefore, taken together, our findings provide evidence for a clear difference in the oral microbiomes between patients with NPC and healthy controls.

Since the relationship between the microbiome and NPC status was strongest in unweighted UniFrac distance, which focuses on presence and absence, we evaluated the relationship between ASV prevalence and disease in a fully adjusted log binomial model adjusted for age, sex, sequencing run, smoking status, community, and the number of missing or repaired teeth. To limit spurious correlations, we defined presence as a relative abundance greater than 0.02% and focused on ASVs present in at least 10% of samples ($n = 245$) (see Fig. S4). We identified 53 ASVs which were significantly different between cases and controls (FDR $P < 0.05$) (Fig. 1e, Table S3). The large majority of these ASVs were more prevalent in controls and came from a wide variety of taxonomic clades, which may suggest a somewhat stochastic loss of ASVs in NPC patients, rather than a systematic loss of specific organisms (see Table S4). This finding is in line with our alpha diversity findings and may indicate overall community instability. In contrast, two ASVs were more prevalent in cases: a member of genus *Lactobacillus* (Lact-eca9) and a *Granulicatella* ASV (Gran-7770).

mSystems®

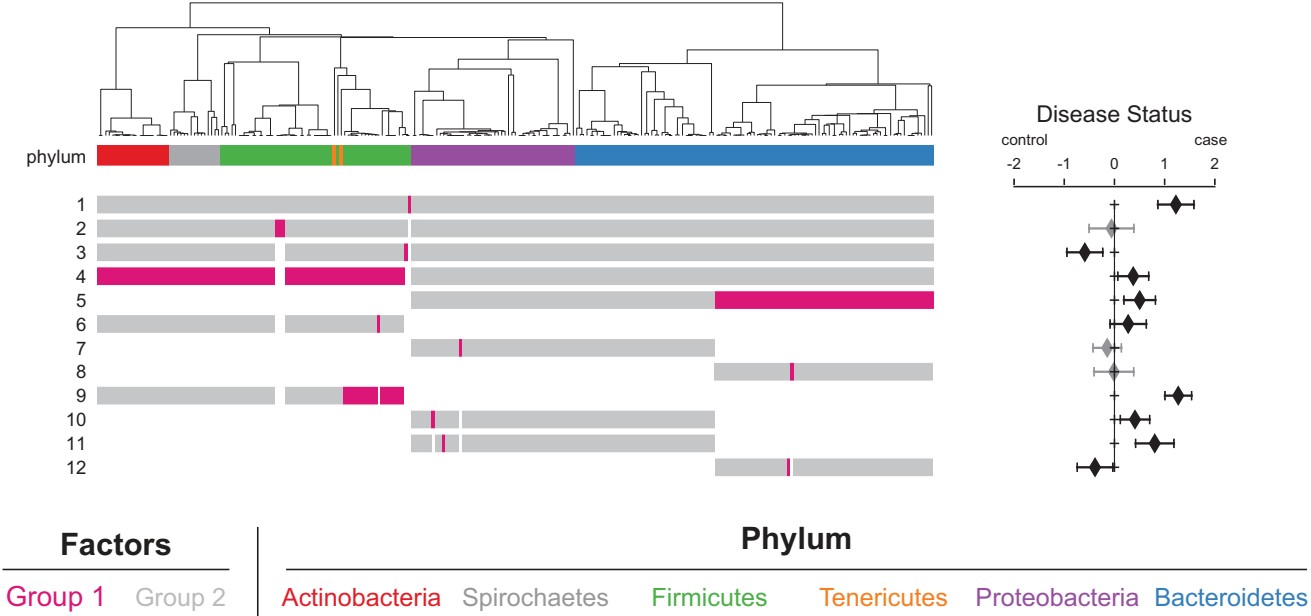

**FIG 2** There are significant associations between phylogenetic partitioning of the taxa and nasopharyngeal carcinoma (NPC) status. The phylogenetic tree with the first 12 Phylofactor-based clade partitions is shown on the left. The top row is colored by phylum; the associated color is shown below. The isometric log transformation is taken as the ratio of the tips highlighted in pink over those highlighted in gray and passed into the regression model to predict the coefficient shown in the forest plot. Clades which are excluded from that factor appear white in the row. The forest plot to the right shows the estimated increase in the factor associated with case-control status based on fitting the ratio in a linear regression adjusted for age, sex, sequencing run, number of missing or repaired teeth, tobacco use, and residential community. Error bars are 95% confidence intervals for the regression coefficient. Black bars indicate significance at *a* of <0.05; gray indicates a nonsignificant association.

To evaluate whether NPC status affected abundance-based partitioning of the microbial community, we applied Phylofactor (31). Our model looked for phylogenetic clades which differentiated NPC cases and controls, adjusting for potential confounders (Fig. 2, Table S4). Of the 12 factors examined, nine were associated with disease status. The primary partition in the data suggested a *Granulicatella* ASV (Gran-7770) was 3.4-fold (95% confidence interval [CI], 2.4- to 4.9-fold) more abundant in NPC cases than in controls. The third factor identified a second *Granulicatella* ASV (Gran-5a37) as less abundant in cases. Both ASVs were also associated with smoking status. We identified three large-scale shifts in microbial abundance associated with NPC status: controls had higher relative abundances of phyla *Proteobacteria* and *Bacteriodetes*, although NPC status was associated with an increased proportion of families *Bacteroidaceae*, *Prevotellaceae*, and contested family *Prevotellaceae* as a portion of their *Proteobacteria* and *Bacteriodetes* ASVs (partitions 4 and 5). Cases also carried a higher relative abundance of class *Bacillus* among the ASVs belonging to class *Clostridia*, class *Erysipelotrichi* (both *Firmicutes*), *Tenericutes*, *Actinobacteria*, and *Spirochetes* (partition 9). The remaining factors associated with NPC status were all single ASVs which differentiated cases and controls, none of which differed in prevalence (Tables S3 and S4).

We complemented the Phylofactor-based analysis of abundance using a modified, multivariate analysis of composition of microbiomes (ANCOM) analysis with the abundant ASVs (32). We identified 6 ASVs with a normalized ANCOM W of at least 0.8, meaning that there was significant change in the ASV compared to the rest of the ASVs in the community in at least 80% of the comparisons (Table S3). All identified ASVs were more abundant in cases. These include the *Granulicatella* ASV (Gran-7770), which Phylofactor identified as the most important feature in the community, two *Streptococcus* (Strep-b566 and Strep-c0f8), the *Lactobacillus* ASV we identified as more prevalent in cases (Lact-eca9), an unspecified member of family *Gemellaceae* (Geme-8a62), and an ASV from genus *Abiotrophia* (Abio-2673). Although only Gran-7770 was identified individually as an important factor in the Phylofactor analysis, all the ASVs we

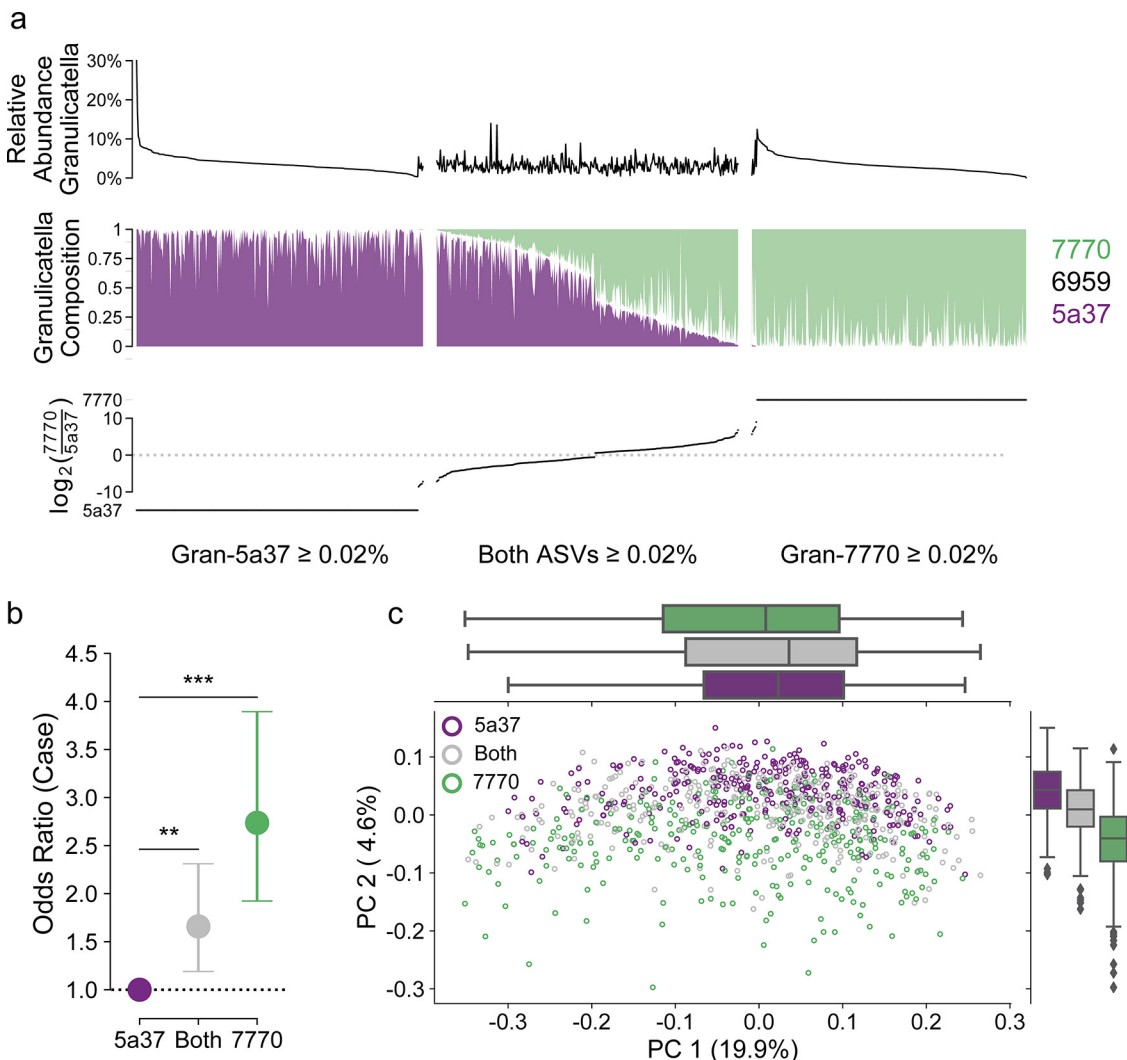

**FIG 3** The *Granulicatella adiacens* variant predicts community structure. (a) *Granulicatella* is a ubiquitous genus, and is present at, on average, 3.2% in all samples. There are three primary ASVs in the genus: Gran-5a37 (purple; *G. adiacens*), Gran-6959 (white; *G. elegans*), and Gran-7770 (green; *G. adiacens*). The ratio between *G. adiacens* variants shows clear separation between the two ASVs. In cases where only one *G. adiacens* ASV was present, we plotted the log ratio as −15 (Gran-5a37) or 15 (Gran-7770) for display purposes. Data are split based on groups assigned using a relative abundance of at least 0.02% as a threshold: Gran-5a37 only, both, or Gran-7770. (b) Nasopharyngeal carcinoma (NPC) cases have significantly higher odds of carrying both Gran-5a37 and Gran-7770 than Gran-5a37 alone and, again, significantly higher odds than carrying either Gran-5a37 and Gran-7770 or Gran-7770. (c) In unweighted UniFrac space, we see separation based on the *G. adiacens* variant along PC2. This figure excludes a single sample which did not contain either Gran-5a37 or Gran-7770.

identified with ANCOM belong to class *Bacillus*, which may explain why it has a higher proportion in Phylofactor.

Based on the significant difference in abundance and prevalence of ASVs from genus *Granulicatella* between cases and controls, we further explored this genus. Two *Granulicatella* species, *Granulicatella adiacens* and *Granulicatella elegans*, are ubiquitous members of the oral microbiome and have been detected across multiple oral environments, although *G. elegans* is frequently localized to the buccal mucosa and keratinized gingiva, while *G. adiacens* localizes to the tongue but occupies a broader niche (33).

Everyone in our study carried at least some *Granulicatella*; on average, the genus represented 3.4% of the relative abundance (Fig. 3a). There was no relationship between the relative abundance of the genus an individual carried and disease status (ANCOM W = 0.25). We identified a total of 14 ASVs in the data set; three were

mSystems®

prevalent enough to be included in our feature-based analyses (Gran-5a37, Gran-7770, and Gran-6959). In 972 (97.8%) individuals, the abundant ASVs were the only *Granulicatella* present. When a BLAST search was performed against the Human Oral Microbiome Database (HOMD), the ASV sequences mapped to two cultured species with more than 99.5% accuracy: *Granulicatella elegans*, which included Gran-6959, and *Granulicatella adiacens* (Gran-7770 and Gran-5a37) (34). Strikingly, we found our two abundant *G. adiacens* ASVs differ by a single nucleotide: Gran-7770 carries a G at nucleotide 119 of our sequence (corresponding approximately to 458 in the full-length 16S rRNA sequence), while Gran-5a37 carries an A.

Gran-7770 was found to be 26% more prevalent among cases, while Gran-5a37 was among the 51 ASVs less prevalent in cases (prevalence ratio [PR], 0.81; 95% CI, 0.74 to 0.88). Both ASVs were also significantly associated with smoking status: Gran-7770 was more prevalent in smokers (PR, 1.48; 95% CI, 1.29 to 1.70), and Gran-5a37 was less prevalent (PR, 0.74; 95% CI, 0.67 to 0.81). There was not a significant relationship between Gran-6959 (*G. elegans*) and either disease status (PR, 0.94; 95% CI, 0.88 to 1.00) or tobacco use (PR, 0.97; 95% CI, 0.90 to 1.06).

We found that 993 of 994 individuals carried at least one *G. adiacens* with a relative abundance of at least 0.02%: 330 (33.2%) carried only Gran-5a37, 316 (31.8%) carried Gran-7770 alone, and 347 (34.9%) carried both (Fig. 3a). For analyses focused on *G. adiacens*, we excluded the single individual who only carried *G. elegans* as an outlier. Among individuals who were classified as carrying only one ASV (Gran-7770 alone or Gran 5a37 alone), the "present" ASV was at least 50-fold more abundant than the other variant (Fig. 3a). In all cases where *G. adiacens* was present, there was at least a 1.5-fold difference between the relative abundances of the two ASVs.

We used a multinomial logistic regression to confirm that disease status was significantly associated with the variants an individual carried: compared to the odds of carrying Gran-5a37 alone, cases had significantly higher odds of carrying both ASVs and, again, significantly higher odds of carrying Gran-7770 alone (Fig. 3b). Although smokers were more likely to have both ASVs or Gran-7770 alone, there was no significant interaction between smoking and disease status.

We also investigated how the presence of a *G. adiacens* variant structured the overall microbial community. We filtered the full ASV table to remove any *Granulicatella* ASVs and used the reduced table to recalculate beta diversity metrics. The *Granulicatella*-free community recapitulated the patterns seen in the full community well (Mantel $R^2$ > 0.91; $P = 0.001$, 999 permutations). We found significant differences between individuals who carried Gran-7770, both, or Gran-5a37 in weighted and unweighted UniFrac and Bray-Curtis distances; all three metrics show clear separation in principal-coordinate analysis (PCoA) space ($P = 0.001$, 999 permutations) (Fig. 3c, Fig. S4). In unweighted UniFrac space (Fig. 3b), the separation was primarily along PC2, likely corresponding to the separation along PC2 seen between cases and controls (Fig. 1d). Furthermore, we found that the *G. adiacens* variant explained 16% of the variation attributed to case-control status in unweighted UniFrac distance and 15% of the variation in weighted UniFrac distance.

We found no relationship between carrying a single *G. adiacens* variant (either Gran-5a37 or Gran-7770 alone) and community richness in the ASV-filtered table, although those who carried both had significantly higher richness ($P = 0.004$). However, the *G. adiacens* variant explained approximately 0.6% of the variation in the observed operational taxonomic units (OTUs). When we incorporated both richness and *G. adiacens* variant into a multivariate Adonis model, we were able to explain 91% of the variance originally attributed to case-control status in unweighted UniFrac distance, 65% in weighted UniFrac. Our results suggest that the *G. adiacens* variant carried by an individual is significantly associated with community structure and may be a route by which NPC status shapes the oral microbiome.

We used a SparCC-based network analysis to identify other community members *Granulicatella* might interact with to exert an effect on the microbiome (35). We were able to identify five networks: one pair of co-occurring ASVs, two pairs of coexcluding

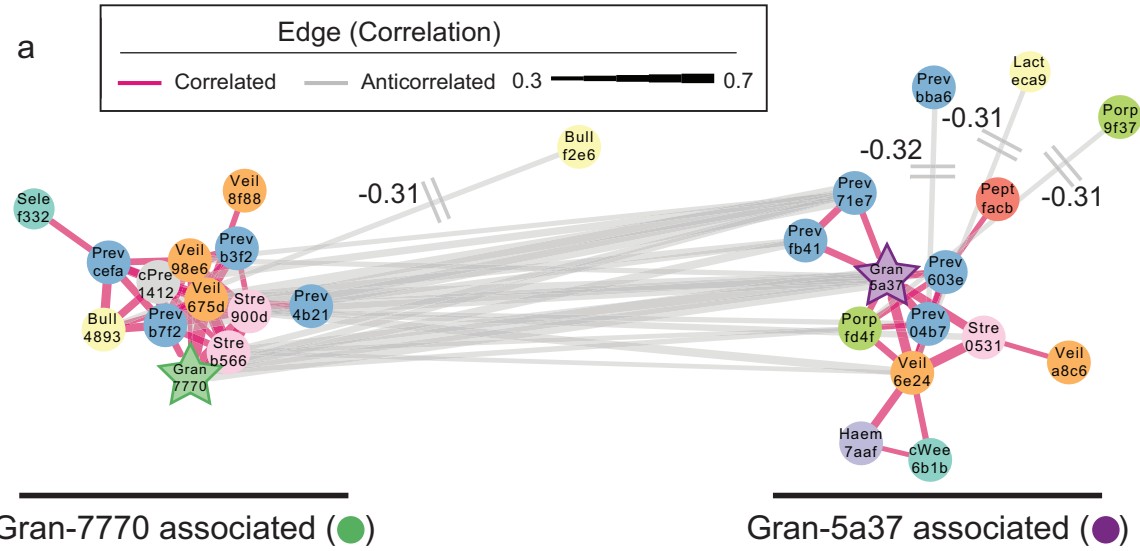

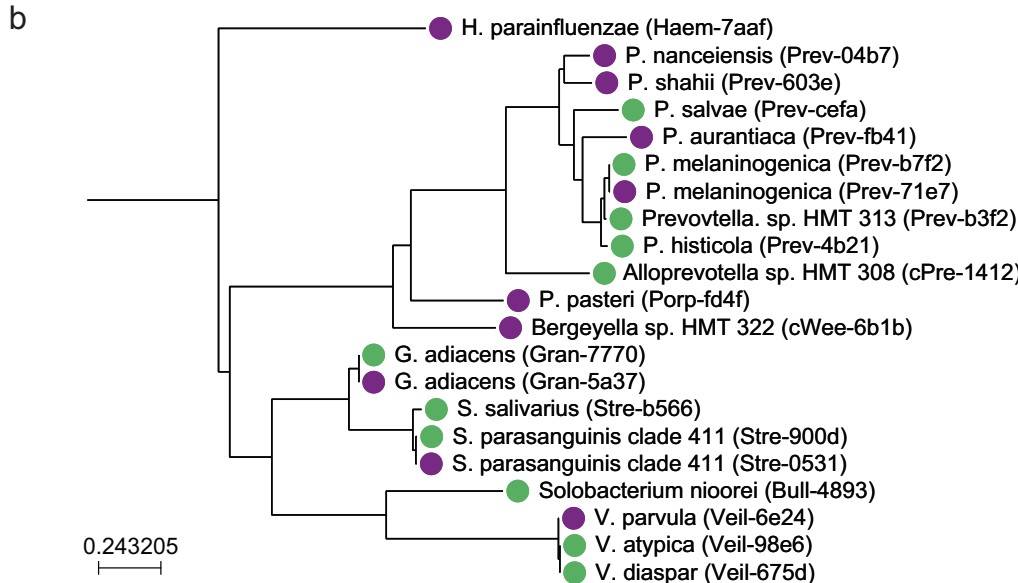

**FIG 4** *Granulicatella adiacens* variants set at the center of a network of closely related co-occurring organisms. (a) SparCC-based network analysis for co-occurring and coexcluding ASVs for all subjects showed a large network with two clusters with common core structures. The color and shape of the nodes are genus specific. The two *G. adiacens* variants are highlighted as stars. The sides of each network are labeled with their associated *G. adiacens* variant. (b) Phylogenetic tree of the core ASVs from the network (positively correlated with either Gran-7770 or Gran-5a37). Tips are labeled by their association with Gran-7770 (green) or Gran-5a37 (purple).

ASVs, one three-member network of co-occurring ASVs, and a large 29-member network of co-occurring and coexcluding ASVs (Fig. 4a). This main network consisted of two clusters of a total of 20 organisms which were positively correlated with a *Granulicatella* variant; the main members of the networks belonged to *Veillonella*, *Streptococcus*, and *Prevotella*. By BLAST searching against HOMD, we identified two additional pairs of ASVs that coexcluded between the two nodes but mapped to the same clones: Stre-900d and Stre-0531 (*Streptococcus parasanguinis* clade 411) and *Prevotella melaninogenica* (Prev-b7f2 and Prev-71e7) (Fig. 4b, Table S5) (34).

We hypothesize the coexcluding networks of ASVs, centered around *Granulicatella*, may reflect partial niche specialization. Previous work suggests quorum sensing networks can form between the core species in our network and that metabolic changes occur in these networks (36, 37). These closely related organisms occupy the same

mSystems®

niches within these metabolic networks; however, strain-specific variation may either respond to or promote disease-associated transformation. Without culture-based experimentation, it is difficult to determine how these organisms may function in concert. One major challenge for *in silico* validation is the limited resolution of existing databases; our results exceed the OTU-based resolution used in construction and span a less frequently characterized hypervariable region.

Within the context of NPC in an endemic region, we hypothesize the oral microbiome may act through several potential mechanisms. The oral microbiome has been suggested to contribute to local tumorigenesis through immune regulation or oncogenic metabolites such as acetaldehyde or nitrosamines (38). An *in silico* study suggested that commercially available strains of *G. adiacens* and coabundant organisms harbor genes involved in nitrate and nitrite reduction (39), although additional work is needed to fully explore the mechanism behind this relationship.

In summary, we have demonstrated a difference in the oral microbial communities between NPC patients and healthy controls in an area of endemicity of southern China, which cannot be explained by other measured factors. The difference is associated with both a loss of community richness and differences among specific organisms, including closely related ASVs from genus *Granulicatella*. In addition, we identified a network of co-occurring and coexcluding ASVs which included these *Granulicatella* variants. These results strongly suggest a relationship between the oral microbiome and nasopharyngeal carcinoma status in newly diagnosed patients.

## MATERIALS AND METHODS

**Study population, survey metadata, and sample collection.** Participant recruitment for the full study was previously described (20). Briefly, incident cases of NPC in Guangdong Provenance and Guangxi Autonomous Region in southern China between March 2010 and December 2013 were invited to participate in the full study. Cases were primarily identified through a rapid ascertainment network at hospitals within the study area and, in some cases, through the Chinese public health network. Age and sex frequency-matched controls were selected from the total population and recruited between November 2010 and November 2014. All study participants had to be between 20 and 74 years of age and had to live in the study region. Individuals with previous malignant disease and acquired or congenital immune deficiency were excluded. Additionally, all individuals had to be fluent in Cantonese and deemed mentally and physically competent to participate. Additionally, controls could not have a diagnosis or history of NPC based on self-reported health history with medical confirmation, and they could not have lived outside the study area for more than 10 years. The study was approved by the institutional review board or ethical review board at all participating centers. All study participants provided written or oral informed consent.

A questionnaire covering demographics and diet, residential, occupational, medical, and family histories was administered in a structured interview. Sample collection occurred at the interview. Participants were asked not to eat or chew gum for 30 min prior to sample collection. Saliva samples (2 ml to 4 ml) were collected into 50-ml falcon tubes with a Tris-EDTA buffer (pH 8.0, 50 mM Tris, 50 mM EDTA, 50 mM sucrose, 100 mM NaCl, 1% SDS) and stored at −20°C not more than 3 days before being stored permanently at −80°C.

For this analysis, we focused on a subset of saliva samples collected within the Wuzhou region of the Guangxi Autonomous Region available at the Wuzhou Red Cross Hospital. There were 1,080 samples submitted for sequencing; two had ambiguous identifiers and could not be mapped to the metadata. Any sample with fewer than 1,000 reads after denoising was excluded, leaving 1,074 saliva samples and 9 negative or single organism controls. Additionally, samples missing information on tobacco use, defined information about tooth brushing frequency, or an undefined residential region ($n = 8$) were excluded (see Fig. S1 in the supplemental material).

Preliminary investigation suggested that the microbial communities for former smokers ($n = 72$) were highly heterogenous (Fig. S2). Sensitivity analyses suggests their exclusion does not alter the major community-level differences. Therefore, they were excluded, leaving a total of 994 individuals in the analysis.

Demographic characteristics and selected covariates of the final study population were compared using a two-sided *t* test for continuous covariates (age) and a chi-square test for categorical covariates. Tests were conducted using scipy 0.19.1 in python 3.5.5 (40).

**DNA extraction, PCR, and sequencing.** Saliva DNA was extracted using a two-step protocol, including the sample preprocessing with lysozyme lysis (lysozyme from chicken egg white; Sigma-Aldrich) and bead beating and the TIANamp blood DNA kit (Beijing, China). The 16S rRNA amplicon library was amplified with 341F/805R primers (CCTACGGGNGGCWGCAG, GACTACHVGGGTATCTAATCC; V3-4) (41, 42). Samples were amplified with 20 cycles of a program with 30 s at 98°C for melting, 30 s at 60°C, and 30 s at 72°C using Hi Fi Kappa HotStart Ready Mix (Roche). Samples were barcoded in a second PCR step (41). DNA cleanup was performed using an Agencourt AMPure XP purification kit. DNA volume and purity were measured on an Agilent 2100 Bioanalyzer system and real-time PCR. Sequencing was

performed at Beijing Genome Institute (BGI) on an Illumina MiSeq using a 2× 300-bp paired-end strategy.

**Denoising, annotation, and filtering.** Samples were demultiplexed using an in-house script. Adaptors were trimmed and paired-end sequences were joined using VSEARCH (v. 2.7) (43). Paired sequences were loaded into the November 2018 release of QIIME 2 (44). Sequences were quality filtered (q2-quality-filter) and denoised using deblur (v. 1.0.4; q2-deblur) with the default parameters on 420-bp amplicons to generate amplicon sequence variants (ASVs) (45, 46). A phylogenetic tree was built using fragment insertion into the August 2013 Greengenes 99% identity tree backbone with q2-fragment-insertion; taxonomic assignments were made with a naive Bayesian classifier trained against the same reference (q2-feature-classifier) (47–49). In cases where the classifier or reference database was unable to describe a taxonomic level (for instance, a missing genus), the taxonomy was described by inheriting the lowest defined level using a custom python script. Following sequencing and denoising, 24,763,933 high-quality reads were retained.

ASVs are identified by the first 4 letters of their lowest taxonomic assignment and the first 4 characters of an MD5 hash of the sequence (Text S1). The full taxonomic assignment and MD5 hashes can be found in Table S3.

**Diversity analyses.** Diversity analyses were performed using samples rarefied to 6,500 sequences. Alpha diversity was calculated as observed ASVs, Shannon diversity, and Faith's phylogenetic diversity using q2-diversity in QIIME 2 (50, 51). Potentially significant alpha diversity categories predictors were identified using a rank sum test in scipy 0.19.1 (40). A $P$ value of 0.05 was considered the threshold for borderline significance for inclusion in a subsequent regression model. Alpha diversity was then evaluated in a multivariate ordinary least-squares (OLS) regression model adjusted for age, sex, and sequencing run number. A final model for each metric was selected by forward selection using models which resulted in decreasing Akaike information criterion (AIC). We checked for the normality of residuals by plotting. The relative contribution of each covariate to that metric was estimated by a "leave-one-out" approach. Regressions were performed in Statsmodels (v. 0.9.0) (52). For visualization, we calculated z-normalized alpha diversity using the mean and standard deviation in diversity for the controls. Alpha diversity was plotted using boxplots in Seaborn 0.9.0 (53, 54).

Beta diversity was measured using the unweighted UniFrac, weighted UniFrac, and Bray-Curtis metrics on rarefied data (q2-diversity) (27–29). Beta diversity was compared using Adonis in the R vegan library (v 2.5-2) adjusted for host age, sex, and sequencing run, with 9,999 permutations (55–57). We used a permdisp test with 999 permutations and the centroid estimate to test for the presence of differences in within-group variation implemented is scikit-bio 0.5.4 (www.scikit-bio.org) (58). Uncorrected $P$ values of less than 0.05 were considered to have significant dispersion, since we were more concerned about false positives than false negatives. Principal-coordinate analyses (PCoAs) were visualized using Emperor (v. 1.0.0b18) and seaborn v. 0.9.0 in matplotlib v. 2.2.3 (53, 59).

In addition to age, sex, and sequencing run, confounders for taxonomic analysis were selected as any covariate with an Adonis $R^2$ at least 60% as large as the value associated with disease in any metric. These included smoking status, residential community, and a categorical description of the number of missing or repaired teeth an individual had.

**ASV regression model.** To look at the relationship between ASV prevalence and disease and smoking status, we used a log binomial regression which was approximated via a Poisson regression with robust standard errors, implemented via base function glm in R and the robust error mechanism implemented via packages lmtest (v 0.9) and sandwich (v. 2.5) in R 3.5 (57, 60–62). The model was adjusted for age, sex, sequencing run, residential community, and the number of missing or repaired teeth. "Presence" was defined as a relative abundance of 1/5,000, which corresponded to the shallowest sequencing depth for the abundant counts. ASVs which were present in more than 1,000 samples were excluded from prevalence analysis. A Benjamini-Hochberg FDR corrected $P$ value of 0.05 was considered statistically significant.

**Phylofactor.** Phylofactor (v. 0.01) was used to look at the relationship between disease status and phylogenetic partitioning between clades (31). Phylofactor is a compositionally aware technique which uses isometric log transforms over an unrooted phylogenetic tree to model differences in the data. This allows the partitioning of data into polyphyletic clades. The Phylofactor multivariate model for each partition was modeled with an ordinary least squares (OLS) regression considering diagnosis, adjusted for residential community, age, sex, number of missing or repaired teeth, tobacco use, and sequencing run. We looked at the first 12 factors using the default parameters, which were optimized for explaining maximal variance. The cladogram and regression coefficient plots were generated in seaborn (53).

**ANCOM.** We applied a modified version of ANCOM 2, implemented in python (33). Briefly, data were offset with a count of 1 to account for the compositional nature, and the ratio of each pair of ASVs was regressed against the disease status, adjusted for age, sex, sequence run, tobacco use, residential community, and the number of missing or repaired teeth using Statsmodels. The W value was calculated the number Benjamini-Hochberg corrected $P$ values associated with the case-control value which were less than 0.05, normalized to the total number of ASVs compared. We set the significance threshold a critical value for W of 0.8 *a priori*.

**Granulicatella.** Total *Granulicatella* was identified by filtering the full ASV table for any ASV assigned to the genus. Species-level assignments were made by using BLAST to search each ASV against the Human Oral Microbiome Database using the online tool; species-level assignments were taken for the cultured species with the best match (34). We treated the abundance of Gran-6959 as the *G. elegans* abundance and the combined abundance of Gran-5a37 and Gran-7770 as the *G. adiacens* abundance

throughout. One sample which did not contain Gran-7770 or Gran-5a37 was excluded from *G. adiacens*-associated analyses.

We used a multinomial logistic regression model, implemented in the nnet library (v. 0.8) in R, to look at whether the carriage of Gran-5a37 alone, Gran-7770 alone, or both ASVs was associated with smoking and disease status (63). The regression was adjusted for age, sex, sequencing run, number of missing or repaired teeth, residential community, the relative abundance of *G. adiacens*, and the relative abundance of *G. elegans*. Having Gran-5a37 was considered the reference group for the multinomial regression.

The effect of *Granulicatella* on alpha and beta diversity was calculated by first filtering out all *Granulicatella* ASVs from the table and then rarifying to 6,250 sequences/sample before diversity calculations. Adonis coefficients were calculated in a model accounting for *G. adiacens* abundance, sequencing run, age, sex, residential community, number of missing or repaired teeth, tobacco use, and disease status. The proportion of disease status explained was calculated by comparing a model excluding the *Granulicatella* variant minus the model including the variant over the model excluding the variant.

**Network analysis.** We used the Sparse Cooccurrence Network Investigation for Compositional data (SCNIC; https://github.com/shafferm/SCNIC) in QIIME 2 (q2-SCNIC) to perform network analysis on the abundant ASVs. The correlation network was built using SparCC, and the network was built using edges with a correlation coefficient of at least 0.3, allowing both co-occurrence and coexclusion (35). Network clusters were identified by finding the most connected node and following all positively correlated nodes in the trimmed SparCC network. Networks were visualized in Cytoscape (v. 3.7.1) using a perfuse-weighted network layout (64). Nodes which were anticorrelated with a single node in the main cluster were trimmed for the sake of visualization; these are labeled with the correlation coefficient.

The phylogenetic tree of core network members was visualized using ete3 (v. 3.1.1) in python 3.6 (65).

**Data availability.** Raw sequencing data and metadata are deposited in ENA under accession PRJEB37445.

## SUPPLEMENTAL MATERIAL

Supplemental material is available online only.

**TEXT S1**, TXT file, 0.1 MB.
**FIG S1**, EPS file, 1.6 MB.
**FIG S2**, JPG file, 0.2 MB.
**FIG S3**, EPS file, 2.6 MB.
**FIG S4**, JPG file, 0.2 MB.
**TABLE S1**, DOCX file, 0.1 MB.
**TABLE S2**, DOCX file, 0.1 MB.
**TABLE S3**, XLSX file, 0.1 MB.
**TABLE S4**, XLSX file, 0.1 MB.
**TABLE S5**, DOCX file, 0.1 MB.

## ACKNOWLEDGMENTS

We thank the study participants, the field work team for the NPCGEE project, the Wuzhou Health System Key Laboratory for Nasopharyngeal Carcinoma Etiology and Molecular Mechanism, and the Key Laboratory of High-Incidence-Tumor Prevention & Treatment (Guangxi Medical University), especially Suhua Zhong and Xiling Xiao, for the processing of salivary samples. The data were stored in the Department of Medical Epidemiology and Biostatistics. We also thank the IT group for their assistance.

This study was funded by the National Cancer Institute of the NIH grant under award number R01CA115873 (principal investigator [PI], H.-O. Adami; co-PI, Y. Zeng), the Swedish Research Council (2015-02625, 2015-06268, and 2017-05814 to W. Ye), the National Natural Science Foundation of China (81272983 to Z. Zhang), and the Guangxi Natural Science Foundation (2013GXNSFGA019002 to Z. Zhang). T. Huang is partly supported by a grant from China Scholarship Council (201408450018).

The study approach was conceived by G.H., H.-O.A., W.Y., Y. Zeng, and Z.Z., A.P., D.B., J.W.D., T.H., W.Y., and Y.C. refined the study design for this project. J.L., Y.C., Y.L., and Y. Zheng were responsible for sample collection and management. D.B. and X.X. performed the lab work, supervised by T.H., X.X., X.Z., and Z.Z., A.P., J.W.D., and T.H. and performed statistical modeling and refinement; bioinformatic and biostatistical analyses were performed by J.W.D. W.Y. contributed to the supervision and coordination of the project. J.W.D. and T.H. wrote the manuscript; A.P. provided critical edits. All authors reviewed and approved the final submission.

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
