## [Reviewer comments · mSystems]

Sub-species niche specialization in the oral microbiome is associated with nasopharyngeal carcinoma risk

Justine Debelius, Tingting Huang, Yonglin Cai, Alexander Ploner, Donal Barrett, Xiaoying Zhou, Xue Xiao, Yancheng Li, Jian Liao, Yuming Zheng, Guangwu Huang, Hans-Olov Adami, Yi Zeng, Zhe Zheng, and Weimin Ye

Corresponding Author(s): Weimin Ye, Karolinska Institutet

Review Timeline:

Submission Date:	January 24, 2020
Editorial Decision:	March 5, 2020
Revision Received:	May 4, 2020
Editorial Decision:	May 8, 2020
Revision Received:	June 17, 2020
Accepted:	June 18, 2020

Editor: Holly Bik

Reviewer(s): Disclosure of reviewer identity is with reference to reviewer comments included in decision letter(s). The following individuals involved in review of your submission have agreed to reveal their identity: Nicholas C K Heng (Reviewer #2)

Transaction Report:

DOI: <https://doi.org/10.1128/mSystems.00065-20>

March 5, 2020

Dr. Weimin Ye
Karolinska Institutet
Stockholm
Sweden

Re: mSystems00065-20 (Sub-species niche specialization in the oral microbiome is associated with nasopharyngeal carcinoma risk)

Dear Dr. Weimin Ye:

The two reviewers were generally very positive about the scope of this study, noting that the manuscript was thorough and well written. However, there were some concerns that microbial ecology concepts were not discussed in enough detail, and the discussion should be expanded to give more context to the study results (see reviewer 2 comments). In addition, all DNA sequences need to be deposited in ENA and the Data Availability paragraph should be updated to include all accession numbers. Furthermore, the authors need to reduce the number of supplemental files - ASM has a limit of 10 supplemental files. I would suggest depositing the extra supplemental material on a data archiving site such as FigShare (<https://figshare.com/>), and providing a link to this external DOI in the Data Availability paragraph.

Below you will find the comments of the reviewers.

To submit your modified manuscript, log onto the eJP submission site at <https://msystems.msubmit.net/cgi-bin/main.plex>. If you cannot remember your password, click the "Can't remember your password?" link and follow the instructions on the screen. Go to Author Tasks and click the appropriate manuscript title to begin the resubmission process. The information that you entered when you first submitted the paper will be displayed. Please update the information as necessary. Provide (1) point-by-point responses to the issues raised by the reviewers as file type "Response to Reviewers," not in your cover letter, and (2) a PDF file that indicates the changes from the original submission (by highlighting or underlining the changes) as file type "Marked Up Manuscript - For Review Only."

Please return the manuscript within 60 days; if you cannot complete the modification within this time period, please contact me. If you do not wish to modify the manuscript and prefer to submit it to another journal, please notify me of your decision immediately so that the manuscript may be formally withdrawn from consideration by mSystems.

To avoid unnecessary delay in publication should your modified manuscript be accepted, it is important that all elements you upload meet the technical requirements for production. I strongly recommend that you check your digital images using the Rapid Inspector tool at <http://rapidinspector.cadmus.com/RapidInspector/zmw/>.

If your manuscript is accepted for publication, you will be contacted separately about payment when the proofs are issued; please follow the instructions in that e-mail. Arrangements for payment must be made before your article is published. For a complete list of **Publication Fees**, including

supplemental material costs, please visit our website.

Sincerely,

Holly Bik

Editor, mSystems

Journals Department
Reviewer comments:

Reviewer #1 (Comments for the Author):

This study utilized 16S rRNA sequencing of DNA from the saliva of nasopharyngeal carcinoma (NPC) cases and sex and aged matched controls to look for significant differences in the oral microbiome. Extensive in silico analysis of the data revealed a reduction in community richness in the cases compared to controls after accounting for potential confounders. Feature-based analysis identified specific amplicon variants in *Granulicatella adiacens* that were strongly associated with NPC status. This is a well-written manuscript. Suggested modifications are listed below.

Page 6, line 111. Please define ASV when first used in the body of the text. Also, since so much of this manuscript is based on ASVs, explain what ASVs are and how they are detected, i.e. what region of the sRNA gene. Why do you look for ASVs anyway?

Page 7, line 140. Figure 1 c,d shows NPC status more important than tobacco use, yet Fig. S3a,b shows tobacco use has a greater effect than NPC status when ASV is taken into account. Either add this to the text here or move S3a,b reference to page 10 when discussing ASV.

Page 11, line 214. What is *Granulicatella*? Overall, is it associated with a health or dysbiosis? How does it fit with the networks identified? What is the relevance to NPC?

Page 13, Survey metadata and sample collection. Does incident mean newly diagnosed, untreated patients? Indicate both inclusion and exclusion criteria, antibiotic usage. Indicate that unstimulated saliva was obtained. How was it stored?

Reviewer #2 (Comments for the Author):

An interesting and well-presented manuscript describing the oral microbiota present in healthy vs NPC patients. The authors also put a case for two putative strains of *Granulicatella adiacens* as centers of niche specialization in NPC patients. However, the rest of the microbiota is not discussed much - how is the NPC microbiome different from the non-NPC state? This manuscript could be improved with more discussion of the oral microbiology aspect.

General comments:

1. Why wasn't dental plaque (subgingival and supragingival) collected and analyzed from all participants? Saliva is more convenient but does not provide data on the microbiota inhabiting specific ecological niches, e.g. the prevalence of *Granulicatella* is usually higher in buccal swabs.
2. What were the relative abundances of Gran-7770 and Gran-5a37 - looks like they were >0.02% but what was the range? Which species (or genus) were most abundant in NPC patients?
3. Were nitrate and nitrite reduction genes present in the genomic DNAs purified from participants? Perhaps real-time PCR could be used for this?

Minor/Specific comments:

1. Lines 188 and 235 - Blasted/Blasting is not a proper verb in this context - "homology searches with BLAST" could be used.
2. Line 268 - 2013 or 2014?
3. Line 276 - composition of Tris-EDTA buffer - concentrations of Tris and EDTA?
4. Line 285 - what PCR enzyme/kit was used to amplify the rRNA genes? What hypervariable region was amplified?
5. Line 289 - Agencourt
6. Line 382-383 - why was the sample that didn't contain both Gran ASVs excluded?
7. Line 432 - accession number not given.
8. Reference list - references 3, 9, 15, 32, 55 and 58 do not have volume/page numbers. Reference 59 has capitalized words in the article title.
9. Line 667 - "... indicates significance at a Benjamini-Hochberg..."

Reviewer #1 (Comments for the Author):

This study utilized 16S rRNA sequencing of DNA from the saliva of nasopharyngeal carcinoma (NPC) cases and sex and aged matched controls to look for significant differences in the oral microbiome. Extensive in silico analysis of the data revealed a reduction in community richness in the cases compared to controls after accounting for potential confounders. Feature-based analysis identified specific amplicon variants in *Granulicatella adiacens* that were strongly associated with NPC status. This is a well-written manuscript. Suggested modifications are listed below.

We thank the reviewer.

Page 6, line 111. Please define ASV when first used in the body of the text. Also, since so much of this manuscript is based on ASVs, explain what ASVs are and how they are detected, i.e. what region of the sRNA gene. Why do you look for ASVs anyway?

Amplicon sequence variants (ASVs) represent a methodological improvement over traditional operational taxonomic units (OTUs) used in microbiome sequencing. Both act as a sort of molecular barcode for organisms and are used in 16s rRNA sequencing as a proxy for organisms which often cannot be resolved with sufficient specificity for genus or species level identification. However, unlike OTUs which are clusters of sequences, ASVs provide single nucleotide resolution. Previous work has suggested single nucleotide resolution is critical to fully understanding the oral microbiome (1), and in the case of our study, this single nucleotide resolution is critical to detecting novel differences in the microbiome.

We have updated the text on page three (lines 111-113) to define amplicon sequence variants and explain their importance:

The oral microbiome was characterized using the V3-4 hypervariable region of the 16s rRNA gene; paired end sequences were denoised to amplicon sequence variants (ASVs). ASVs provide a molecular barcode with single nucleotide resolution to distinguish different organisms.

Page 7, line 140. Figure 1 c,d shows NPC status more important than tobacco use, yet Fig. S3a,b shows tobacco use has a greater effect than NPC status when ASV is taken into account. Either add this to the text here or move S3a,b reference to page 10 when discussing ASV.

We thank the reviewer for their careful reading. This reflects the use of different metrics between the two figures to provide a more robust understanding of the community. All our distance metrics are calculated based on ASVs, and the difference is that unweighted UniFrac (Figure 1c) considers only the presence/absence while both weighted UniFrac and Bray Curtis distance (figure S3) account for abundance. Most larger studies identify a difference in the oral microbiome between smokers and non-

smokers measured with beta diversity; however, the studies are heterogenous and multiple metrics are rarely used (2–5).

Lines 150-154 mention tobacco use as a factor in the beta diversity analysis, as did lines 163-165.

Page 11, line 214. What is *Granulicatella*? Overall, is it associated with a health or dysbiosis? How does it fit with the networks identified? What is the relevance to NPC?

Granulicatella is a ubiquitous oral genus. It was originally described as a nutritionally variant *Streptococcus*, but molecular phylogeny indicates it is a unique genus. Two species live within the human oral cavity: *G. elegans* and *G. adicans*. *G. elegans* is localized to the buccal mucosa, while *G. adicans* is often found on the dorsal tongue but has a less strict habitat (6). We have presented a similar summary in lines 224-228.

Much of the medical literature related to the genus describes it as an opportunistic pathogen (7). Previous studies have reported a decrease in *Granulicatella* in periodontal disease (although due to the age of the studies, it is somewhat unclear if this is the consequence of failing to account for compositionality) (8, 9); in a very small number of asthmatics (10). It is also reported to be associated with metabolic syndrome in a Korean study (11). However, these cases differ from our results in that we see no association between the abundance or prevalence of the genus as a whole and disease status (line 231-232), instead it is the specific ASV-level differences that are associated with outcome.

We cannot find literature which associates *Granulicatella* with head and neck cancers nor in previous descriptions of the oral microbiome in NPC, suggesting ours is a novel finding. There is also little literature around its interaction with other co-occurring species in the network. Previous papers have described biofilms containing *Veillonella*, *Streptococcus* and *Granulicatella* (12, 13), however, since this is a novel finding, we are unsure of its implications in this context.

Page 13, Survey metadata and sample collection. Does incident mean newly diagnosed, untreated patients? Indicate both inclusion and exclusion criteria, antibiotic usage. Indicate that unstimulated saliva was obtained. How was it stored?

Many of these details for the full cohort were described in the original publication describing the full study cohort (14). We have expanded lines 330 – 343 to describe inclusion criteria and definitions:

Participant recruitment for the full study has been previously described (20). Briefly, incident cases of NPC in Guangdong Provenance and Guangxi Autonomous Region in southern China between March 2010 and December 2013 were invited to participate in the full study. Cases were primarily identified through a rapid ascertainment network at hospitals within the study area and in some cases through the Chinese public health

network. Age and sex frequency-matched controls were selected from the total population and recruited between November 2010 and November 2014. All study participants had to be between 20 and 74 years of age and had to live in the study region. Individuals with previous malignant disease and acquired or congenital immune deficiency were excluded. Additionally, all individuals had to be fluent in Cantonese and deemed mentally and physically competent to participate. Additionally, controls could not have a diagnosis or history of NPC based on self-reported health history with medical confirmation, nor could they have lived outside the study area for more than 10 years. The study was approved by the Institutional Review Board or Ethical Review Board at all participating centers. All study participants provided written or oral informed consent.

The original study was designed starting in 2008 and sample collection began in 2010. Information about antibiotic use was not collected (lines 169-176). However, there is some evidence that the oral microbiome is resilient to antibiotic use and so although we recognize this as a limitation we do not believe that it alters our conclusions.

Reviewer #2 (Comments for the Author):

An interesting and well-presented manuscript describing the oral microbiota present in healthy vs NPC patients. The authors also put a case for two putative strains of *Granulicatella adiacens* as centers of niche specialization in NPC patients. However, the rest of the microbiota is not discussed much - how is the NPC microbiome different from the non-NPC state? This manuscript could be improved with more discussion of the oral microbiology aspect.

We thank the reviewer for their comments and careful reading.

General comments:

1. Why wasn't dental plaque (subgingival and supragingival) collected and analyzed from all participants? Saliva is more convenient but does not provide data on the microbiota inhabiting specific ecological niches, e.g. the prevalence of *Granulicatella* is usually higher in buccal swabs.

We agree with the reviewer that this would indeed be an interesting approach, especially in light of our novel finding with *G. adicans*. However, these samples come from a larger epidemiological study (15) and were originally collected to sequence the human and Epstein Barr Virus genomes (i.e. Xu et al (16)). Perhaps in light of our results, a future study will focus on this area.

2. What were the relative abundances of Gran-7770 and Gran-5a37 - looks like they were >0.02% but what was the range?

For all the abundant ASVs in our study, we defined "present" as at least 1/5000 sequences in a sample. We picked this threshold based on rarefaction depth after filtering and it was selected without consideration for specific organisms.

We added a line describing the average abundance of *Granulicatella* in the samples (line 230-232). We have also expanded figure 3 and added an additional set of panels (3a) to provide a more comprehensive descriptive view of the total *Granulicatella* abundance, the relative abundance of each ASV variant, and the ratio between Gran-7770 and Gran-5a37.

Which species (or genus) were most abundant in NPC patients?

We introduced a descriptive paragraph (lines 120-126) describing the overall observed salivary microbiome. We also expanded the description of the phylofactor result (lines 201-206) and introduced ANCOM to test abundance-based differences at the ASV-level (lines 210-221). We chose to use ASVs here rather than collapse the data to genus since ASVs are more likely to be externally valid if other researchers choose to use different databases. We also believe that the single nucleotide resolution granted by ASVs is superior to the collapsed genus-level information and better recapitulates the underlying community. We also applied ANCOM collapsed to genus level, but nothing

achieved the a priori threshold of a normalized W of 0.8. We have chosen not to present these results in the text other than specifically in relation to *Granulicatella* (line 231-232).

3. Were nitrate and nitrite reduction genes present in the genomic DNAs purified from participants? Perhaps real-time PCR could be used for this?

We agree this is an interesting experiment and perhaps we will be able to conduct it in the future. However, the effect of nitrate/nitrite reducing genes is one of multiple hypotheses for the mechanism behind the relationship with the oral microbiome and NPC presented in lines 311-316.

Minor/Specific comments:

1. Lines 188 and 235 - Blasted/Blasting is not a proper verb in this context - "homology searches with BLAST" could be used.

We have corrected these lines.

2. Line 268 - 2013 or 2014?

Cases were between 2010 and 2013; controls between 2010 and 2014. We have expanded the methods section to address this.

3. Line 276 - composition of Tris-EDTA buffer - concentrations of Tris and EDTA?
50mM Tris and 50mM EDTA were used. This has been added in lines 348-350.

4. Line 285 - what PCR enzyme/kit was used to amplify the rRNA genes? What hypervariable region was amplified?

We have added detailed information in the "DNA extraction, PCR, and sequencing" section (lines 370-379).

5. Line 289 - Agencourt

6. Line 382-383 - why was the sample that didn't contain both Gran ASVs excluded?

There was a single sample which did not contain either variant. Our analyses focused on comparing the effect of *G. adicans* carriage specifically, and this single sample reflected a unique carriage state (neither ASV). It is not possible to characterize a distribution from a single sample and not possible to perform statistical tests without a distribution. The sample was therefore excluded as an outlier specifically from the *G. adicans* related analyses, although it is present in all other sections, including the co-occurrence networks. We have expanded the description in figure 3 to describe this exclusion and explained it in the text.

7. Line 432 - accession number not given.

We have added the accession number: PRJEB37445. Data will be made publicly available upon publication.

8. Reference list - references 3, 9, 15, 32, 55 and 58 do not have volume/page numbers. Reference 59 has capitalized words in the article title.

We have added volume and page number to 3, 9, and 15 and updated the title in 59. Reference 32 is a book.

Reference 55 and 58 are software packages/releases. The versions we used do not have corresponding publications and therefore no volume or page number.

9. Line 667 - "... indicates significance at a Benjamini-Hochberg..."

References

1. Eren AM, Borisy GG, Huse SM, Mark Welch JL. 2014. Oligotyping analysis of the human oral microbiome. *Proc Natl Acad Sci U S A* 111.
2. Mason MR, Preshaw PM, Nagaraja HN, Dabdoub SM, Rahman A, Kumar PS. 2015. The subgingival microbiome of clinically healthy current and never smokers. *ISME J* 9:268–72.
3. Wu J, Peters BA, Dominianni C, Zhang Y, Pei Z, Yang L, Ma Y, Purdue MP, Jacobs EJ, Gapstur SM, Li H, Alekseyenko A V, Hayes RB, Ahn J. 2016. Cigarette smoking and the oral microbiome in a large study of American adults. *ISME J* 10:2435–46.
4. Morris A, Beck JM, Schloss PD, Campbell TB, Crothers K, Curtis JL, Flores SC, Fontenot AP, Ghedin E, Huang L, Jablonski K, Kleerup E, Lynch S V., Sodergren E, Twigg H, Young VB, Bassis CM, Venkataraman A, Schmidt TM, Weinstock GM. 2013. Comparison of the respiratory microbiome in healthy nonsmokers and smokers. *Am J Respir Crit Care Med* 187:1067–1075.
5. Vallès Y, Inman CK, Peters BA, Ali R, Wareth LA, Abdulle A, Alsafar H, Anouti F Al, Dhaheri A Al, Galani D, Haji M, Hamiz A Al, Hosani A Al, Houqani M Al, Junaibi A Al, Kazim M, Kirchhoff T, Mahmeed W Al, Maskari F Al, Alnaeemi A, Oumeziane N, Ramasamy R, Schmidt AM, Weitzman M, Zaabi E Al, Sherman S, Hayes RB, Ahn J. 2018. Types of tobacco consumption and the oral microbiome in the United Arab Emirates Healthy Future (UAEHFS) Pilot Study. *Sci Rep* 8:11327.
6. Mark Welch JL, Dewhirst FE, Borisy GG. 2019. Biogeography of the Oral Microbiome: The Site-Specialist Hypothesis. *Annu Rev Microbiol* 73:335–358.
7. Cargill JS, Scott KS, Gascoyne-Binzi D, Sandoe JAT. 2012. Granulicatella infection: diagnosis and management. *J Med Microbiol* 61:755–61.
8. Liu B, Faller LL, Klitgord N, Mazumdar V, Ghodsi M, Sommer DD, Gibbons TR, Treangen TJ, Chang Y-C, Li S, Stine OC, Hasturk H, Kasif S, Segrè D, Pop M, Amar S. 2012. Deep sequencing of the oral microbiome reveals signatures of periodontal disease. *PLoS One* 7:e37919.
9. Colombo AP V, Boches SK, Cotton SL, Goodson JM, Kent R, Haffajee AD, Socransky SS, Hasturk H, Van Dyke TE, Dewhirst F, Paster BJ. 2009. Comparisons of subgingival microbial profiles of refractory periodontitis, severe periodontitis, and periodontal health using the human oral microbe identification microarray. *J Periodontol* 80:1421–32.
10. Wang L, de Ángel Solá D, Mao Y, Bielecki P, Zhu Y, Sun Z, Shan L, Flavell RA, Bazy-Asaad A, DeWan A. 2018. Family-based study reveals decreased abundance of sputum Granulicatella in asthmatics. *Allergy* 73:1918–1921.
11. Si J, Lee C, Ko G. 2017. Oral Microbiota: Microbial Biomarkers of Metabolic Syndrome Independent of Host Genetic Factors. *Front Cell Infect Microbiol* 7:516.
12. Chalmers NI, Palmer RJ, Cisar JO, Kolenbrander PE. 2008. Characterization of a Streptococcus sp.-Veillonella sp. Community Micromanipulated from Dental Plaque. *J Bacteriol* 190:8145–8154.

13. Palmer RJ, Diaz PI, Kolenbrander PE. 2006. Rapid succession within the *Veillonella* population of a developing human oral biofilm in situ. *J Bacteriol* 188:4117–24.
14. Ye W, Chang ET, Liu Z, Liu Q, Cai Y, Zhang Z, Chen G, Huang Q-H, Xie S-H, Cao S-M, Shao J-Y, Jia W-H, Zheng Y, Liao J, Chen Y, Lin L, Liang L, Ernberg I, Vaughan TL, Huang G, Zeng Y, Zeng Y-X, Adami H-O. 2017. Development of a population-based cancer case-control study in southern china. *Oncotarget* 8.
15. Ye W, Chang ET, Liu Z, Liu Q, Cai Y, Zhang Z, Chen G, Huang Q-H, Xie S-H, Cao S-M, Shao J-Y, Jia W-H, Zheng Y, Liao J, Chen Y, Lin L, Liang L, Ernberg I, Vaughan TL, Huang G, Zeng Y, Zeng Y-X, Adami H-O. 2017. Development of a population-based cancer case-control study in southern china. *Oncotarget* 8:87073–87085.
16. Xu M, Yao Y, Chen H, Zhang S, Cao S-M, Zhang Z, Luo B, Liu Z, Li Z, Xiang T, He G, Feng Q-S, Chen L-Z, Guo X, Jia W-H, Chen M-Y, Zhang X, Xie S-H, Peng R, Chang ET, Pedergnana V, Feng L, Bei J-X, Xu R-H, Zeng M-S, Ye W, Adami H-O, Lin X, Zhai W, Zeng Y-X, Liu J. 2019. Genome sequencing analysis identifies Epstein-Barr virus subtypes associated with high risk of nasopharyngeal carcinoma. *Nat Genet* 51:1131–1136.

May 8, 2020

Dr. Weimin Ye
Karolinska Institutet
Stockholm
Sweden

Re: mSystems00065-20R1 (Sub-species niche specialization in the oral microbiome is associated with nasopharyngeal carcinoma risk)

Dear Dr. Weimin Ye:

I am satisfied that the authors have addressed all remaining reviewer concerns, and I am now happy to recommend this manuscript for publication at mSystems. However, before final acceptance please ensure that the underlying data in the "Data Availability" section is made public. The ENA accession number (PRJEB37445) is not yet publicly available - please ensure these data are published before final acceptance .

Below you will find the comments of the reviewers.

To submit your modified manuscript, log onto the eJP submission site at <https://msystems.msubmit.net/cgi-bin/main.plex>. If you cannot remember your password, click the "Can't remember your password?" link and follow the instructions on the screen. Go to Author Tasks and click the appropriate manuscript title to begin the resubmission process. The information that you entered when you first submitted the paper will be displayed. Please update the information as necessary. Provide (1) point-by-point responses to the issues raised by the reviewers as file type "Response to Reviewers," not in your cover letter, and (2) a PDF file that indicates the changes from the original submission (by highlighting or underlining the changes) as file type "Marked Up Manuscript - For Review Only."

Due to the SARS-CoV-2 pandemic, our typical 60 day deadline for revisions will not be applied. I hope that you will be able to submit a revised manuscript soon, but want to reassure you that the journal will be flexible in terms of timing, particularly if experimental revisions are needed. When you are ready to resubmit, please know that our staff and Editors are working remotely and handling submissions without delay. If you do not wish to modify the manuscript and prefer to submit it to another journal, please notify me of your decision immediately so that the manuscript may be formally withdrawn from consideration by mSystems.

To avoid unnecessary delay in publication should your modified manuscript be accepted, it is important that all elements you upload meet the technical requirements for production. I strongly recommend that you check your digital images using the Rapid Inspector tool at <http://rapidinspector.cadmus.com/RapidInspector/zmw/>.

Sincerely,

Holly Bik

Editor, mSystems

Journals Department
Reviewer comments:

June 18, 2020

Prof. Weimin Ye
Karolinska Institutet
Stockholm
Sweden

Re: mSystems00065-20R2 (Sub-species niche specialization in the oral microbiome is associated with nasopharyngeal carcinoma risk)

Dear Prof. Weimin Ye:

I have now confirmed that the authors have made all associated datasets publicly available, as requested in the final round of revision.

Your manuscript has been accepted, and I am forwarding it to the ASM Journals Department for publication. For your reference, ASM Journals' address is given below. Before it can be scheduled for publication, your manuscript will be checked by the mSystems senior production editor, Ellie Ghatineh, to make sure that all elements meet the technical requirements for publication. She will contact you if anything needs to be revised before copyediting and production can begin. Otherwise, you will be notified when your proofs are ready to be viewed.

Sincerely,

Holly Bik
Editor, mSystems

Journals Department
Table S1: Accept
Figure S2: Accept
Table S3: Accept
Figure S1: Accept
Table S4: Accept
Figure S4: Accept
File S1: Accept
Table S2: Accept
Figure S3: Accept
Table S5: Accept